# Solving the Green Open Vehicle Routing Problem Using a Membrane-Inspired Hybrid Algorithm

**Yunyun Niu** [1], **Zehua Yang** [1], **Rong Wen** [2], **Jianhua Xiao** [3] and **Shuai Zhang** [4,*]

1   School of Information Engineering, China University of Geosciences in Beijing, Beijing 100083, China; yniu@cugb.edu.cn (Y.N.); zhyang1997@163.com (Z.Y.)
2   Singapore Institute of Manufacturing Technology, Singapore 138634, Singapore; wenr@simtech.a-star.edu.sg
3   The Research Center of Logistics, Nankai University, Tianjin 300071, China; jhxiao@nankai.edu.cn
4   DeGroote School of Business, McMaster University, Hamilton, ON L8S 4M4, Canada
*   Correspondence: zhans72@mcmaster.ca

**Abstract:** The green open vehicle routing problem with time windows has been widely studied to plan routes with minimal emissions in third-party logistics. Due to the NP-hardness, the performance of the general heuristics significantly degrades when dealing with large-scale instances. In this paper, we propose a membrane-inspired hybrid algorithm to solve the problem. The proposed algorithm has a three-level structure of cell-like nested membranes, where tabu search, genetic operators, and neighbourhood search are incorporated. In particular, the elementary membranes (level-3) provide extra attractors to the tabu search in their adjacent level-2 membranes. The genetic algorithm in the skin membrane (level-1) is designed to retain the desirable gene segments of tentative solutions, especially using its crossover operator. The tabu search in the level-2 membranes helps the genetic algorithm circumvent the local optimum. Two sets of real-life instances, one of a Chinese logistics company, Jingdong, and the other of Beijing city, are tested to evaluate our method. The experimental results reveal that the proposed algorithm is considerably superior to the baselines for solving the large-scale green open vehicle routing problem with time windows.

**Keywords:** membrane computing; P system; open vehicle routing problem; carbon emission; tabu search

## 1. Introduction

The vehicle routing problem (VRP) introduced by Dantzig and Ramser in 1959 [1] is most commonly studied for route planning in logistics. It is defined as the determination of the routes along which a fleet of freight vehicles fulfills the needs of a set of customers (or nodes) at various locations, with the objective of optimising the total cost. Thereafter, a number of VRP variants have been proposed and investigated [2]. Among them, the vehicle routing problem with time windows (VRPTW) has been widely explored, as more practical factors are considered [3]. In this problem, each customer has to be served within their own prescribed time interval, which is named the time window constraint [4].

The open vehicle routing problem with time window (OVRPTW) is a variant of the VRPTW [5]. Compared to the VRPTW, vehicles in the OVRPTW are not required to return to the depot after fulfilling the delivery. This phenomenon is quite popular in many real-world scenarios, where vehicles may not need to return to the depot if they directly finish after serving the customers, especially when a third-party company is delivering. Considering the benefits of higher operation efficiency and the resource utilisation rate, outsourcing freight shipping to third-party logistics companies may have considerable cost savings. Currently, due to the extensive concern about environmental pollution, the logistics industry needs to reduce greenhouse gas emissions [6] to achieve green and sustainable transportation. The OVRPTW is further extended to the green OVRPTW (GOVRPTW), i.e., the objective of the studied problem is to minimise the total greenhouse gas emissions.

The first solution to the open vehicle routing problem (OVRP) was proposed by Bodin et al. [7] in 1983. In that study, a variant of the Clarke and Wright algorithm was designed to develop open routes for airplanes. From then on, various heuristics and metaheuristics have been developed to solve the OVRP. The most widely used are based on a search algorithm, e.g., the tabu search [8], the neighbourhood-based search [9], and the threshold accepting algorithm [10]. Meanwhile, bioinspired and population-based metaheuristics have also been developed, such as the particle swarm optimisation [11], the ant colony optimisation [12], and genetic and evolutionary computing [13].

Recently, Ashtineh and Pishvaee [14] evaluated the economic and environmental impacts of alternative fuels in the VRP through measurement and quantification of the effects for the emitted pollutant. Yu et al. [15] considered a heterogeneous fleet of vehicles to reduce carbon emissions in the green vehicle routing problem with time windows (GVRPTW). Wang and Lu [16] presented a memetic algorithm with competition (MAC) to solve the capacitated green vehicle routing problem. We investigated the fuel consumption in route planning while considering the third-party logistics company, which was formulated based on a comprehensive modal emission model [17]. Specifically, a hybrid tabu search algorithm integrated with several neighbourhood search strategies was leveraged to solve this problem. Although desirable performance was achieved in [17], there is still much room for improvement, especially given that the GOVRPTW is of strong NP-hardness, where the computation becomes prohibitively intractable as the problem scales up.

As a promising branch of the bioinspired intelligent optimisation approach [18], membrane computing has been identified as an effective distributed and parallel model, which is also known as the P system [19]. The theory and applications of the P system and its variants provide a theoretical possibility to solve NP-complete problems in polynomial time [20] and have been widely studied in various fields [21], such as ecosystem and pedestrian behaviour, engineering computing and optimising, and so on. Motivated by those successful applications, we expect that more effective optimisation algorithms could be derived based on the P system for solving GOVRPTW, especially when integrated with evolutionary algorithms. In this paper, we exploit a membrane-inspired hybrid heuristic algorithm to solve the large-scale GOVRPTW to improve the performance over [17], which was based on a hybrid tabu search. The main contributions of this paper are summarised as follows.

(1) A novel three-level nested membrane structure is designed with respective algorithms. To be specific, the skin membrane acts as the first level, where a genetic algorithm is mainly exploited to search for solutions to the routing problems. Six adjacent inner membranes act as the second level, where different tabu search algorithms are exploited to find tentative solutions. The elementary membrane in each level-2 membrane acts as the third level, where neighbourhood search operations are exploited to facilitate adjusting the search direction of the corresponding level-2 membrane.

(2) Communication channels between the level-2 membranes and their inner membranes are designed to exchange solutions to favourably find better solutions. Communication channels also exist between the skin membrane and the level-2 membranes, where the crossover operator in the genetic operators is leveraged to retain satisfactory gene segments. In addition, the tabu search with different attractors is adopted to help the genetic algorithm escape from the local optimum. The convergence curve cliffs after each communication justify the effectiveness of the communication channels.

(3) Experiments are carried out on large-scale real-world problem instances, i.e., a Beijing 100-nodes set and a Jingdong 1000-nodes instance. The results demonstrate that our method significantly outperformed the hybrid tabu search [17], tabu search, and genetic algorithm, respectively. In particular, the computation time observed when comparing the performance on the Jingdong 1000-nodes instances and the Beijing 100-nodes instances further demonstrates the superiority of our algorithm in solving large-scale problems.

## 2. Related Work

### 2.1. Algorithms for the OVRP

Various heuristics and metaheuristics have been developed to solve the OVRP. There is much research on tabu search, neighbourhood-based search, and the threshold accepting algorithm. Brando [8] presented a novel tabu search algorithm for the open vehicle routing problem. Derigs et al. [22] proposed an attribute-based hill climber heuristic, which was a parameter-free variant of the tabu search principle. Fu et al. [23] presented a new tabu search heuristic for finding the routes that minimised two objectives while satisfying three constraints. Russell et al. [24] proposed a tabu search metaheuristic to aid in the coordination and synchronisation of the production and delivery of multiproduct newspapers to bulk delivery locations. Fleszar et al. [9] proposed an effective variable neighbourhood search heuristic. Pisinger et al. [25] presented a unified heuristic with an adaptive large neighborhood search framework to solve five different variants of the VRP. Salari et al. [26] proposed a heuristic improvement procedure based on integer linear programming techniques. Zachariadis et al. [27] presented an innovative local search metaheuristic, which examined wide solution neighbourhoods. Tarantilis et al. proposed an annealing-based method that utilised a backtracking policy [28] and a single-parameter metaheuristic method that exploited a list of threshold values to intelligently guide an advanced local search [10].

Population-based metaheuristics have also been proposed. MirHassani et al. [11] presented a real-value version of particle swarm optimisation for solving the OVRP. Wang et al. [29] proposed a novel real number encoding method of particle swarm optimisation. Zhen et al. [30] proposed a novel particle swarm optimisation in which the vehicle was mapped into the integer part of the real number. Li et al. presented an ant colony system hybridised with local search [31] and an ant colony optimisation-based metaheuristic [12]. Pan et al. [32] presented a clonal selection algorithm. Repoussis et al. [13] proposed a hybrid evolution strategy. Yu et al. [33] applied a novel hybrid algorithm combining the genetic algorithm and the tabu search. The tabu search can help the genetic algorithm circumvent the local optimum. In our previous work, a hybrid tabu search algorithm was proposed to minimise the fuel consumption of the OVRPTW [17]. However, the performance of the algorithms mentioned above always degrades when dealing with a large-scale instance.

### 2.2. Membrane Algorithms

Membrane computing is a branch of natural computing. It is inspired by the structure and the function of living cells, tissues, and organs. It provides a distributed and parallel framework for modelling and high-performance computation. Barbuti et al. [34] proposed minimal probabilistic P systems as modelling notation for ecological systems. Lucie et al. [35] summarised the most important results on P colonies. Niu et al. [36] proposed a simulation model called an intelligence decision P system inspired by the process of cell migration. Sakellariou et al. [37] used a population P system in the agent-based simulation modelling of passengers boarding an underground station.

Nishida T. Y. [38] proposed the first membrane-inspired algorithm and proved its efficiency in solving the travelling salesman problem (TSP). Zhang et al. [39] analysed and optimised radar emitter signals by leveraging membrane algorithms. An optimisation spiking neural P system was presented to approximately solve the general combinatorial optimisation problems [40]. Zhang et al. [41] proposed a population–membrane-system-inspired evolutionary algorithm, in which a population P system and a quantum-inspired evolutionary algorithm were used. Membrane algorithms adopt rich and varied frameworks, which facilitate the cooperation of multiple algorithms. It helps to design a hybrid algorithm and use the respective advantages of different algorithms for large-scale NP-hard problems. In this work, a novel three-level membrane algorithm was designed for the large-scale GOVRPTW instances. It can also be considered an extension of the application field of membrane algorithms.

### 3. Problem Formulation

The problem studied in this paper takes into account practical factors such as vehicle fuel consumption, greenhouse gas emissions, third-party logistics, and time window constraints on the basis of the classical vehicle routing problem, which can be modelled as a green open vehicle routing problem with time window (GOVRPTW). The GOVRPTW could be defined on a complete and directed graph $\mathcal{G} = (N, A)$, where $N$ is the node set and $A$ is the arc set. In particular, $N = \{0, \ldots, n\}$ includes $n+1$ entities, with 0 representing the depot and $N_0 = N \setminus \{0\}$ representing the customer set. Each customer $i$ has a positive demand $q_i$. The arc set $A = \{(i, j) : i, j \in N, i \neq j, j \neq 0\}$ represents the connection between the nodes. The goods delivery is considered as opposed to the goods pick-up problem. The demand of any customer $q_i$ is assumed to be less than the vehicle capacity $Q$. In this work, it is assumed that the traffic conditions on all roads are uncongested or free-flow such that vehicle speeds can be optimised. The vehicles will finish after completing their service to the customers rather than returning to the depot. The notations used in the problem description are summarised in Notations part. We adopt the comprehensive emissions model developed by Barth et al. [42], Barth et al. [43], and Scora et al. [44] to estimate actual fuel consumption and gas emissions. The objective function of the GOVRPTW is formulated as follows [17].

$$Minimise \sum_{(i,j)\in A} \lambda f_c k N_e V d_{ij} \sum_{r=1}^{R} z_{ij}^r / v_{ij}^r \tag{1}$$

$$+ \sum_{(i,j)\in A} \lambda f_c \gamma \alpha_{ij} d_{ij} (w x_{ij} + f_{ij}) \tag{2}$$

$$+ \sum_{(i,j)\in A} \lambda f_c \beta \gamma d_{ij} \sum_{r=1}^{R} (v_{ij}^r)^2 z_{ij}^r \tag{3}$$

$$+ \sum_{j\in N_0} f_d s_j, \tag{4}$$

where $\lambda = \xi / \kappa \psi$, $\gamma^h = 1/1000 n_{tf} \eta$ and $\alpha = \tau + g\sin\theta + gC_r\cos\theta$ are constants; $\beta = 0.5 C_d \rho A$ is a vehicle-specific constant; the values of the parameter used in the formulation are given in Table 1, and the reader can refer to Koc et al. [45] for more details. The length of arc $(i, j) \in A$ is denoted by $d_{ij}$; the total weight of a vehicle on arc $(i, j)$ is calculated as $w + f_{ij}$, with $w$ being the weight of a vehicle and $f_{ij}$ being the amount of freight flow on arc $(i, j) \in A$; the binary variable $x_{ij}$ equals 1 if a vehicle travels along the arc $(i, j) \in A$, and it is 0 otherwise; and the binary variable $z_{ij}^r$ equals 1 if a vehicle $r$ ($r = 1, 2, \ldots$) travels along the arc $(i, j) \in A$ at speed $v_{ij}^r$, and it is 0 otherwise. The objective is to minimise the total cost of three components: the fuel consumption, the $CO_2$ emissions, and the total wage of drivers. The cost induced by the fuel consumption and $CO_2$ emissions is represented by the first three terms in the objective. Specifically, term (1) describes the engine module cost, term (2) computes the weight module cost, and term (3) reflects the speed module cost. The cost of the driver wage is represented by the fourth term in the objective. The constraints of the GOVRPTW are shown as follows.

$$\sum_{j\in N_0} x_{0j} \leq |N_0|, \tag{5}$$

$$\sum_{i\in N} x_{ij} = 1, \ \forall j \in N_0 \tag{6}$$

$$\sum_{j\in N} x_{ij} \leq 1, \ \forall i \in N_0 \tag{7}$$

$$\sum_{i=1}^{n} x_{i0} = 0, \tag{8}$$

$$\sum_{j \in N} f_{ij} - \sum_{j \in N} f_{ji} = q_i, \ \forall i \in N_0 \tag{9}$$

$$q_j x_{ij} \le f_{ij} \le (Q - q_i) x_{ij}, \ \forall (i, j) \in A \tag{10}$$

$$y_i - y_j + t_i + \sum_{r=1}^{R} d_{ij} z_{ij}^r / v_{ij}^r \le M_{ij}(1 - x_{ij}), \ \forall i \in N, \ j \in N_0, \ i \ne j \tag{11}$$

$$a_i \le y_i \le b_i, \ \forall i \in N_0 \tag{12}$$

$$\sum_{r=1}^{R} z_{ij}^r = x_{ij}, \ \forall (i, j) \in A \tag{13}$$

$$x_{ij} \in \{0, 1\}, \ \forall (i, j) \in A \tag{14}$$

$$z_{ij}^r \in \{0, 1\}, \ \forall (i, j) \in A, \ r = 1, \ldots, R \tag{15}$$

$$f_{ij} \ge 0, \ \forall (i, j) \in A \tag{16}$$

$$y_i \ge 0, \ \forall i \in N_0 \tag{17}$$

$$s_i \ge 0, \ \forall i \in N_0 \tag{18}$$

where $t_i$ is the time consumed for serving customer $i \in N_0$; $[a_i, b_i]$ is the time window of customer $i \in N_0$, within which the customer must be visited; $y_j$ is the actual starting time for serving node $j \in N_0$; if a vehicle arrives at a customer before its $a_i$, it has to wait because it can only start service at or after $a_i$; $s_j$ is the total time cost of the route in which the last visited customer is $j \in N_0$. Moreover, constraint (5) defines the maximum number of vehicles; constraints (6)–(8) ensure that every customer is served only once, and the vehicles finish after delivery rather than returning to where they started; constraints (9) and (10) together define the freight flows; the time window is described in constraints (11) and (12), in which $M_{ij} = max\{0, b_i + t_i + d_{ij}/v_{ij}^r - a_j\}$; constraint (13) imposes that each arc has only one speed level; and constraints (14)–(18) define the range of the variables.

**Table 1.** Parameter values of a light-duty vehicle.

| Notation | Description | Typical Value |
|:---:|:---:|:---:|
| $\eta$ | Diesel engine efficiency | 0.45 |
| $C_r$ | Rolling resistance coefficient | 0.01 |
| $\rho$ | Density of air (kg/m$^3$) | 1.2041 |
| $\psi$ | Conversion factor (g/s to litre/s) | 737 |
| $\kappa$ | Heating value of the typical diesel fuel (kilojoule/g) | 44 |
| $g$ | Constant of gravitation (m/s$^2$) | 9.81 |
| $n_{tf}$ | Vehicle drive train efficiency | 0.45 |
| $\xi$ | Fuel-to-air mass ratio | 1 |
| $\tau$ | Acceleration (m/s$^2$) | 0 |
| $\theta$ | Angle of the road | 0 |
| $v^u$ | Highest speed (m/s) | 27.8 (or 100 km/h) |
| $v^l$ | Lowest speed (m/s) | 5.5 (or 20 km/h) |
| $f_c$ | Cost of fuel and CO$_2$ emissions (GBP/litre) | 1.4 |
| $f_d$ | Cost of driver wage (GBP/s) | 0.0022 |

**Table 1.** *Cont.*

| Notation | Description | Typical Value |
| --- | --- | --- |
| $Q$ | Vehicle capacity (kg) | 4000 |
| $w$ | Vehicle curb weight (kg) | 3500 |
| $f$ | Vehicle fixed cost (GBP/day) | 0 |
| $V$ | Engine displacement (litre) | 4.5 |
| $k$ | Engine friction factor (kilojoule/rev/litre) | 0.25 |
| $N_e$ | Engine speed (rev/s) | 38.34 |
| $A$ | Area of frontal surface (m$^2$) | 7.0 |
| $C_d$ | Aerodynamics drag coefficient | 0.6 |

## 4. The Proposed Method

In this section, a membrane-inspired hybrid heuristic algorithm is proposed to deal with the GOVRPTW. As depicted in Figure 1, the MIHA has three levels of cell-like nested membranes. To be specific, the skin membrane constitutes the first level. The second level consists of six adjacent inner membranes (labelled as 1, . . . , 6). The elementary membrane in each level-2 membrane constitutes the third level. The membranes of level-2 can provide the tentative solutions to the skin membrane through unidirectional channels, while the communication channels between the level-2 membranes and the corresponding level-3 membranes are bidirectional. In the membranes of level-3, the neighbourhood search operations are leveraged to help adjust the search direction of the corresponding level-2 membranes. On the one hand, the GA operators in the skin membrane, especially the crossover operator, are exploited to retain the desired gene segments. On the other hand, the tabu search algorithms with different attractors in the level-2 membranes are leveraged to help the GA algorithm escape from the local optimum.

The general framework of the MIHA is introduced in Algorithm 1. In the initialisation stage, the preliminary solutions are generated by using six different operators. Afterwards, a speed improvement strategy is exploited to determine the optimal speed on each route in the solution. In this case, the initial population would be generated in the skin membrane. The searching processes in the level-2 membranes are guided by the tabu search algorithm, while the evolution in the skin membrane proceeds according to the genetic operators. After every $I_{MCA}$ steps, the archive solutions of each level-2 membrane would be transferred to the skin membrane to help update the current population. When the termination condition is met, the final output is defined as the best solution found from all different membranes.

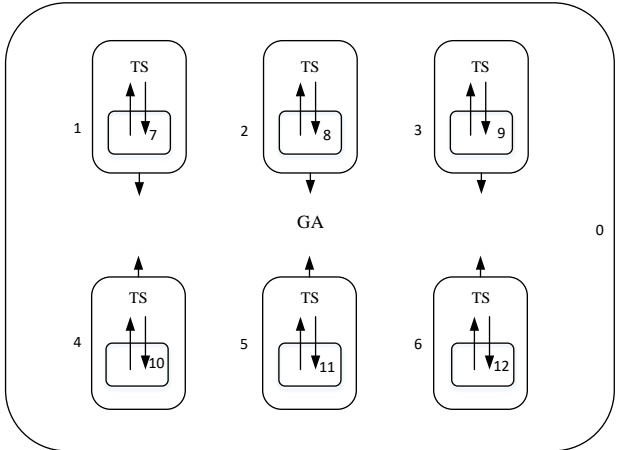

**Figure 1.** The proposed membrane structure.

---

**Algorithm 1** Pseudo-code of MIHA

---

**Require:** maximum number of iterations $I_{max}$, iteration number before MCA $I_{MCA}$
**Ensure:** $x_{best}$.

  1: **for** $m = 1$ to 6 **do**
  2:      Generate initial solution $x_{initial}^m$ by using operator $m$
  3:      Send $x_{initial}^m$ to level-2 membrane $m$
  4:      Set $x_{current}^m = x_{initial}^m$
  5: **for** $i = 1$ to $I_{max}$ **do**
  6:      **for** $m = 1$ to 6 **do**
  7:          $\{x_{neighbors}\}$ = level2_search($x_{current}^m$)
  8:          update($archive^m$, $\{x_{neighbors}\}$)
  9:          Send $\{x_{neighbors}\}$ to the adjacent $level3\_membrane$
 10:          $\{x_{neighbors}\}$ = level3_search($\{x_{neighbors}\}$)
 11:          update($archive^m$, $\{x_{neighbors}\}$)
 12:          **if** $archive^m \setminus tabu\_list^m \neq \varnothing$ **then**
 13:              Set $x_{current}^m$ = the_best_of($archive^m \setminus tabu\_list^m$)
 14:              update($tabu\_list^m$, $x_{current}^m$)
 15:      **if** $i$ mod $I_{MCA} == 0$ **then**
 16:          MCA($skin\_membrane$, $level2\_membrane^{1-6}$)
 17: Output the best solution $x_{best}$

---

The encoding approach of the solution has a significant impact on the quality of the final result as well as the computational efficiency. In our method, a complete solution consists of a number of routes, and the variable-length chromosomes [46] are adopted to encode the routes, where a chromosome comprising the integer nodes represents a route. Specifically, each vehicle departs from the first node and ends at the last node it serves; the travel speed $v_{ij}^r$ between every two adjacent points $i$ and $j$ on the same path is to be decided; the service start time $y_i$ of each point $i$ can be calculated; a solution is feasible only when it does not violate any constraints.

*4.1. Initialisation*

During initialisation, six unique operators were adopted to create different initial solutions as shown in Figure 2. Then, the population of the skin membrane and the archive solutions of the level-2 membranes were formed. Specifically, the six operators were executed based on the following heuristics (or rules). Moreover, the pseudo-code is given in Algorithm 2.

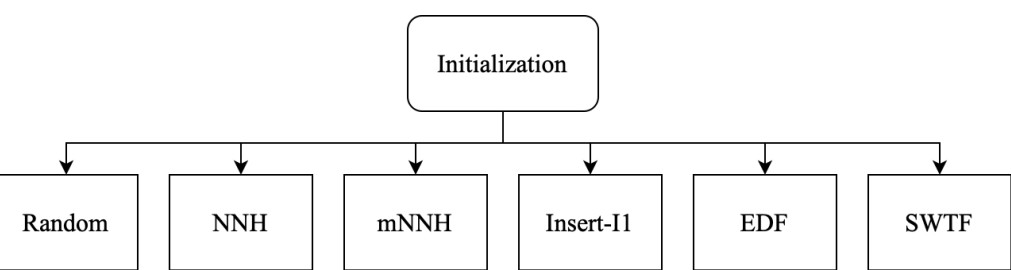

**Figure 2.** Six operators for initialising solutions.

---

**Algorithm 2** Pseudo-code of initialisation

---

**Require:** Operators: Random, NNH, mNNH, $I_1$, EDF, SWTF
**Ensure:** Implement the initialisation of MIHA
 1: operators = $\{Random, NNH, mNNH, I_1, EDF, SWTF\}$
 2: **for** $m$ = 1 to 6 **do**
 3:     $x_{initial}^m$ = create_initial_solution(operator[m])
 4:     Send $x_{initial}^m$ to *level2_membrane$^m$*
 5:     $\{x_{neighbors}^m\}$ = level2_search($x_{initial}^m$)
 6:     Send $\{x_{neighbors}^m\}$ to *skin_membrane*

---

(1)  Random heuristic: It randomly chooses routes that satisfy constraints (5)–(17).
(2)  Nearest neighbourhood heuristic (NNH): It generates a set of routes according to the distance from the current node. The nearest customer to the depot is chosen as the start node $x_0$ of the first route. Then it chooses an unassigned customer $x_1$, who is nearest to $x_0$. It repeats the same procedure until no feasible candidate nodes for the current route can be found. This also means that the current route is completed. Then, it allocates a new vehicle for the next route and constructs the route in a similar way until all customer nodes are assigned.
(3)  Modified nearest neighbourhood heuristic (mNNH): It generates a set of routes according to both the demand of the next customer and the distance from the current node. During the shipping process, a vehicle can offload $q_i$ payload after servicing customer $i$. The unit distance payload of customer $j$ on arc $(i, j)$ is defined as $\overline{\Delta f_{ij}} = q_j/d_{ij}$. The mNNH creates a number of routes sequentially by considering the $\overline{\Delta f_{ij}}$ as an objective. First, it chooses the customer $c$ that satisfies $c = argmax_{c \in N_{ucs}}\{|\overline{\Delta f_{0c}}|\}$, where $N_{ucs}$ represents the unassigned customer set, which includes the initial current node of the first route. Next, the feasible customer $nc$ that satisfies $nc = argmax_{nc \in N_{ucs}}\{|\overline{\Delta f_{c,nc}}|\}$ is selected as the next node $c$ and added to the current route. Then, it repeats choosing the next node and appending it to the current route. If no more feasible nodes can be added, the current route is completed, and another route will be constructed in a similar way until all customers have been assigned.
(4)  Insert $I_1$ heuristic: As first proposed by Solomon [47], the customer $u^*$ is chosen based on the Equations (19) and (20) and then inserted to the route according to the insert $I_1$ heuristics. Moreover, the feasible and desired position of the selected $u^*$ in the route is decided by Equations (21) and (22) as follows, where $y_{j_u}$ is the new time for service to begin at customer $j$, given that $u$ is on the route. The main idea is to use several criteria to insert a new customer into the current partial route at every iteration.

$$c_2(i^*, u^*, j^*) = maximum[c_2(i, u, j)], \tag{19}$$

$$c_2(i, u, j) = \lambda d_{0u} - c_1(i, u, j), \ \lambda \geq 0 \tag{20}$$

$$c_1(i^*, u, j^*) = min[c_1(i, u, j)], \tag{21}$$

$$c_1(i, u, j) = \alpha_1(d_{iu} + d_{uj} - \mu d_{ij}) + \alpha_2(y_{j_u} - y_j), \ \mu, \alpha_1, \alpha_2 \geq 0, \ \alpha_1 + \alpha_2 = 1. \tag{22}$$

(5)  Earliest deadline first heuristic: It selects the customer with the earliest (or tightest) deadline for service at each step.
(6)  Shortest waiting time first heuristic: It selects the customer with the shortest waiting time.

*4.2. GA in Skin Membrane*

After the initialisation, the population in the skin membrane evolved according to the rationale of GA. To select parent chromosomes for the crossover operator, the binary tournament was implemented, while the chromosomes for the mutation operators were randomly selected.

This process was repeated several times to obtain sufficient parent chromosomes. Route-exchange crossover [46] was used to retain the better gene segment. The routes in one solution chromosome were reproduced and shared with others. In order to satisfy the constraints, the duplicated nodes in a chromosome were removed if a new route was inserted into it. The single point mutation was adopted as the mutation operator in the skin membrane.

### 4.2.1. Crossover Operator

The performance of the genetic algorithm is highly affected by the crossover and mutation operators. Using these operators, the search space can be more effectively explored and better solutions can be exploited. In the literature of the genetic algorithm, many crossover and mutation operators have been developed for different optimisation problems. In our method, the route-exchange crossover proposed by [46] was adopted to retain the desirable gene segment. Different from the classical one-point crossover, which may produce infeasible route sequences, the route-exchange crossover operator allows the favourable sequences of routes or the genes in a chromosome to be shared with other chromosomes in the evolving population. Firstly, we selected parents to perform the crossover operation by binary tournament. Secondly, the route-exchange crossover was performed on the selected parents, in which the best routes of the selected chromosomes were exchanged. To ensure the feasibility of chromosomes after the crossover, duplicated customers were deleted from the original routes, while the newly inserted route was left unchanged.

### 4.2.2. Mutation Operator

We adopted three mutation operators, i.e., random, split-longest, and merge-shortest operators [46] to extend the search space. The corresponding operators are listed as follows.

(1) Random: This operation randomly removes a customer node from a given route and inserts it into another feasible position of the origin route.
(2) Split-longest: This operation searches for the route with the highest total cost and breaks the route into two parts at a random point.
(3) Merge-shortest: This operation searches for the two routes of the chromosome with the smallest total cost and appends one to the other.

### 4.3. Tabu Search in Level-2 Membranes

After the initialisation, the tabu search was started in each level-2 membrane as described in Figure 3 and Algorithm 3. First, the current solution was used to create the neighbour solutions. If any neighbour solution was better than the solutions in the archive, it replaced the most inferior one. Next, the level-3 membrane performed the neighbourhood search and the corresponding result helped the archive to find better solutions. Then, the best one was selected as the next solution from the archive rather than from the tabu list. Finally, the current solution was added into the tabu list.

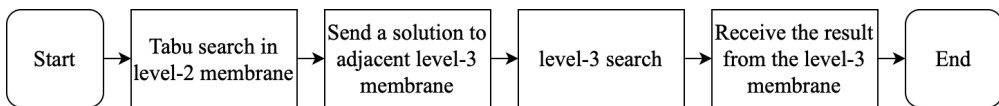

**Figure 3.** Level-2 membrane search.

Pertaining to the neighborhood search, three operators were exploited.

(1) Random operator: It randomly exchanges the position of two nodes of a given solution, provided that no constraint is violated.
(2) High-cost-node operator: It removes a high-cost customer node defined as $u^* = argmax_{u \in N}\{d_{iu} + d_{uj}\}$, where $i$ is the preceding customer and $j$ is the succeeding customer, and inserts the node into another position.
(3) Long-wait-time operator: It relocates the customer with a long wait time node defined as $u^* = argmax_{u \in N}\{a_u - e_u\}$, where $e_u$ is the arrival time of customer $u$.

---

**Algorithm 3** Level-2 Membrane Search Algorithm

---

1: $\{x_{neighbors}\}$ = search_neighbors($x_{current}$)
2: update_its_archive($\{x_{neighbors}\}$)
3: Send $\{x_{neighbors}\}$ to the adjacent *level3_membrane* for level-3 search
4: **if** *archive* $\setminus$ *tabu_list* $\neq \varnothing$ **then**
5: 　　Set $x_{current}$ = the_best_of(*archive* $\setminus$ *tabu_list*)
6: 　　update_its_tabu_list($x_{current}$)

---

### 4.4. Neighbourhood Search in Level-3 Membranes

In the level-3 membrane, the neighbourhood search was performed with a specific probability, which is denoted as $p_{level3}$, to find the superior solutions and improve the diversity of the archive solutions, as illustrated in Figure 4 and Algorithm 4. First, it randomly selected a solution from the archive of the adjacent level-2 membrane. Next, it used this solution as an input to perform the neighbourhood search. Finally, it sent the output of the neighbourhood search back to its adjacent level-2 membrane. If the output solutions were better than the ones in the archive of the level-2 membrane, they replaced the inferior ones. This search procedure is expected to find superior solutions as well as improve the diversity of archive solutions.

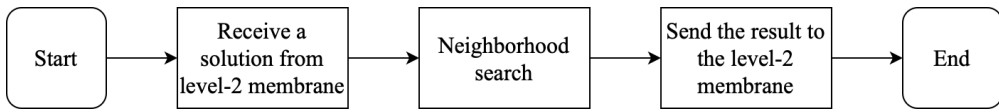

**Figure 4.** Level-3 membrane search.

---

**Algorithm 4** Level-3 Membrane Search Algorithm

---

1: Randomly select a solution from the archive of adjacent *level2_membrane*, as $x_{random}$
2: Send $x_{random}$ to current *level3_membrane*
3: $\{x_{neighbors}\}$ = search_neighbors($x_{random}$, $p_{level3}$)
4: Send $\{x_{neighbors}\}$ back to the *level2_membrane*
5: update(*archive$_{level2\_membrane}$*, $\{x_{neighbors}\}$)

---

### 4.5. Communications between the Level-2 Membrane and Skin Membrane

After a specific number of iterations each time, archive solutions in each level-2 membrane were transported to the skin membrane, and the current population was updated. We denote this number of iterations as $I_{MCA}$ and this communication as Membrane Communication Algorithm (MCA) (described in Figure 5 and Algorithm 5). First, the archive solutions of each level-2 membrane were merged with current individuals in the skin membrane. Next, the best $P$ solutions were selected and combined to form a new population. Specifically, the GA operators implemented in the skin membrane, especially the crossover operators, were used to retain the desirable gene segments of solutions found by the tabu search in the level-2 membrane. Various solutions obtained by the tabu search algorithms with different attractors facilitated the GA escaping from the local optimal solutions.

---

**Algorithm 5** Membrane Communication Algorithm

---

1: **for** $m$ = 1 to 6 **do**
2: 　　Send archive solutions of *level2_membrane*$[m]$ to *skin_membrane*
3: Select the best $P$ solutions in *skin_membrane*
4: Rebuild the population of *skin_membrane* with those solutions

---

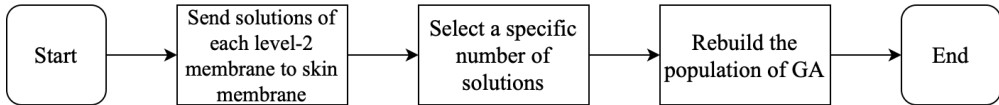

**Figure 5.** Membrane communication.

### 4.6. Speed Optimisation

The speed of the vehicle in each arc has a significant impact on the fuel consumption and the total cost. The process of determining the optimal speed of each route in a solution is important to minimise the fuel consumption costs and driver wages. After obtaining a solution that comprised a number of routes, we implemented the speed and departure-time optimisation (SD-TOA) proposed by Karmer et al. [48] to compute the optimal speed for each route. As shown in Algorithm 6, the SD-TOA was executed based on a divide-and-conquer strategy. To be specific, the whole route was divided into several subroutes by first ignoring the time window constraints, and then the sub-routes were re-optimised recursively. If a resulting subroute satisfied all time window constraints, then it was returned. Otherwise, the customer with the maximum time-window violation was identified and its arrival time was set to the closest feasible value. Fixing this time window failure created two subroutes, which were re-optimised recursively.

---

**Algorithm 6** Speed and Departure-time Optimisation Algorithm (SDTOA)

---

1: Procedure $SDTOA(s, e)$
2: $p \leftarrow violation \leftarrow maxViolation \leftarrow 0$
3: $D \leftarrow \sum_{i=s}^{e-1} d_{i,i+1}$
4: $T \leftarrow \sum_{i=s}^{e-1} \tau_i$
5: **if** $s = 1$ **and** $e = n_\sigma$ **then**
6:      $t_1 = a_1$
7: **if** $e = n_\sigma$ **then**
8:      $t_e = min\{max\{a_e, t_s + D/v_{FD}^* + T\}, b_e\}$
9: **if** $s = 1$ **then**
10:      $t_s = min\{max\{a_s, t_e - D/v_{FD}^* - T\}, b_s\}$
11: $v_{REF} \leftarrow D/(t_e - t_s - T)$
12: **for** $i = s + 1 \ldots e$ **do**
13:      $t_i = t_{i-1} + \tau_{i-1} + d_{i-1,i}/v_{REF}$
14:      $violation = max\{0, t_i - b_i, a_i - t_i\}$
15:      **if** $violation > maxViolation$ **then**
16:          $maxViolation = violation$
17:          $p = i$
18:      **if** $maxViolation > 0$ **then**
19:          $t_p = min\{max\{a_p, t_p\}, b_p\}$
20:          $SDTOA(s, p)$
21:          $SDTOA(p, e)$
22:      **if** $s = 1$ **and** $e = n_\sigma$ **then**
23:          **for** $i = 2 \ldots n_\sigma$ **do**
24:              $v_{i-1,i} = max\{d_{i-1,i}/(t_i - t_{i-1} - \tau_{i-1}), v_F^*\}$

---

## 5. Computational Results

In this section, we evaluate the proposed MIHA on two sets of real-life data set. The presented algorithm was implemented in Matlab on a PC with an Intel Core i5-10400 processor, 16G RAM, and Microsoft Windows 10 operating system. The parameters used in our algorithm are listed in Table 2. We first test the MIHA on Beijing instance set, where the customer locations were scattered in both urban and suburban areas, and realistic geographical road information was leveraged to calculate the relevant costs [17]. Then experiments are conducted on larger-scale Jingdong instances. Twenty independent experiments were

conducted for each instance. Experimental results are listed in Tables 3–10, where the columns represent the best solution, the mean solution, the worst solution, the standard deviation (SD), and the elapsed time (ET, also known as computation time), respectively.

*5.1. Parameter Analysis*

*5.1.1. $p_{level3}$*

In the level-3 membrane, an extra neighbourhood search was implemented with a specific probability $p_{level3}$ to help find better solutions and improve the diversity of solutions. Here, we conducted the experiments on the Beijing example with 60 (customer) nodes, i.e., BJ60_01, to analyse the influence of parameter $p_{level3}$, the results of which are recorded in Table 3. According to Table 3, either too low or too high probability deteriorated the performance. We suggest that lower probability was less effective in finding better solutions, while higher probability might lead the whole evolution to undesirable areas. Considering that the best overall result was captured at $p_{level3} = 0.8$, we used this value in the subsequent experiments.

**Table 2.** Algorithm parameters.

| Notation | B | Typical Values |
|:---:|:---:|:---:|
| $I_{MCA}$ | Iterations between MCAs | 150 |
| $I_{max}$ | Maximum iteration number | 500 |
| $p_{level3}$ | Probability of level-3 search | 0.8 |
| $Ar$ | Archive size | 100 |
| $Ns$ | Neighbourhood size | 100 |
| $L$ | Tabu-list size | 30 |
| $P$ | Population size | 100 |
| $p_{mutation}$ | Rate of mutation | 0.8 |
| $p_{crossover}$ | Rate of crossover | 0.2 |
| $(\alpha_1, \alpha_2, \mu, \lambda)$ | $I_1$ parameters | $(0.5, 0.5, 1, 1)$ |

**Table 3.** Experimental result with different $p_{level3}$.

| $p_{level3}$ | Best Solution | Mean Solution | Worst Solution | SD | ET |
|:---:|:---:|:---:|:---:|:---:|:---:|
| 0 | 10,875.6553 | 10,928.4416 | 10,957.5933 | 22.3405 | 98.4386 |
| 0.1 | 10,882.5664 | 10,935.8356 | 10,978.9560 | 25.6220 | 99.7357 |
| 0.2 | 10,897.0627 | 10,922.4866 | 10,946.9486 | 14.2856 | 106.7122 |
| 0.3 | 10,894.6855 | 10,935.1849 | 10,981.7598 | 22.6840 | 114.1316 |
| 0.4 | 10,893.8999 | 10,927.5347 | 10,960.1311 | 20.8581 | 132.3853 |
| 0.5 | 10,870.5658 | 10,927.1250 | 10,965.2899 | 28.0809 | 139.0837 |
| 0.6 | 10,893.7139 | 10,922.8795 | 10,946.2359 | 19.7116 | 153.5391 |
| 0.7 | 10,859.4377 | 10,918.7540 | 10,956.2807 | 23.5941 | 158.2609 |
| 0.8 | 10,849.7328 | 10,907.9905 | 10,959.0818 | 28.7783 | 172.7458 |
| 0.9 | 10,885.8673 | 10,922.2359 | 10,945.8091 | 16.2176 | 172.7154 |
| 1 | 10,864.7334 | 10,914.4491 | 10,939.8273 | 21.2864 | 179.3093 |

*5.1.2. $I_{MCA}$*

As described previously, after a specific number of iterations each time, i.e., $I_{MCA}$, the archive solutions of each level-2 membrane were sent to the skin membrane to help update the population. Here, the impact of parameter $I_{MCA}$ is discussed. We still leveraged the same instance, i.e., BJ60_01, to evaluate this parameter. As displayed in Table 4, the best result was captured at $I_{MCA} = 150$, which means that sending archive solutions of level-2 membranes to the skin membrane every 150 iterations achieved better performance.

Similarly, according to these results, either a smaller or larger $I_{MCA}$ may deteriorate the solution; hence, we used $I_{MCA}$ = 150 in the subsequent experiments.

**Table 4.** Experimental results with different $I_{MCA}$.

| $I_{MCA}$ | Best Solution | Mean Solution | Worst Solution | SD | ET |
|---|---|---|---|---|---|
| 15 | 10,887.9339 | 10,915.9258 | 10,933.9978 | 12.4846 | 180.9297 |
| 25 | 10,887.3386 | 10,907.1512 | 10,946.9533 | 17.9590 | 179.3292 |
| 50 | 10,864.7334 | 10,914.4491 | 10,939.8273 | 21.2864 | 179.3093 |
| 75 | 10,887.7475 | 10,914.0151 | 10,947.0650 | 18.9695 | 175.5044 |
| 100 | 10,899.9133 | 10,917.2956 | 10,930.3250 | 9.1312 | 174.3383 |
| 125 | 10,881.1171 | 10,913.0205 | 10,956.5405 | 19.0231 | 180.1102 |
| 150 | 10,847.0901 | 10,900.6532 | 10,947.3101 | 26.0596 | 150.6687 |
| 175 | 10,866.3854 | 10,908.5777 | 10,944.5708 | 23.6946 | 158.3528 |
| 200 | 10,878.1705 | 10,911.7143 | 10,945.2829 | 18.0546 | 184.6337 |

*5.2. Effectiveness of Search in Level-3 Membranes*

In this subsection, the effectiveness of the search in level-3 membranes is analyzed, where experiments were conducted on the Beijing 60-node instances with light-duty vehicles. The tabu search algorithm elicited desirable solutions faster by obtaining the attractor from the level-3 membranes. The MIHA without the level-3 membranes is denoted as MIHA$_{-level3}$ for convenience. The performance of MIHA and MIHA$_{-level3}$ on our example is shown in Table 5. It is easily observed that the level-3 membranes demonstrated considerable advantages in finding solutions with a lower total cost.

*5.3. Effectiveness of Tabu Search*

The tabu search algorithm plays a significant role in the level-2 membrane. In order to analyse the effectiveness of the tabu search algorithm, this part of the experiment used the greedy algorithm to replace the tabu search algorithm, i.e., MIHA$_{-TS}$, and compared the experimental results with MIHA. Although the membrane framework was retained in MIHA$_{-TS}$, the demonstrated performance was less competitive than those achieved using MIHA, as shown in Table 6.

*5.4. Effectiveness of GA in Skin Membrane*

The crossover operator of the genetic algorithm can combine the excellent gene fragments of different individuals, and the mutation operator can help expand the search space. As an important part of the skin membrane, it is necessary to analyse the effectiveness of the genetic algorithm in the skin membrane. In MIHA, the solutions transmitted by the level-2 membranes formed the initial population of the skin membrane and finally output the excellent feasible solutions through the crossover and mutation operations in the genetic algorithm. In this subsection, MIHA without the genetic algorithm is denoted as MIHA$_{-GA}$, in which the ability to integrate gene fragments from different membranes was absent, and the output was the best solutions from the level-2 membranes. Table 7 records the experimental results of both MIHA and MIHA$_{-GA}$, respectively. The GA in the skin membrane had a favourable impact on the performance of our algorithm, as it achieved solutions with lower costs. Regarding the proposed MIHA, various solutions obtained by the tabu search of level-2 membranes with different attractors helped the GA escape from the local optimum. As a result, the cliffs were observed every $I_{MCA}$ iterations in the convergence curve of the skin membrane, as depicted in Figure 6, which further justified the superiority of our design.

**Table 5.** Comparison of MIHA and MIHA$_{-level3}$.

|  | Instance | Best Solution | Mean Solution | Worst Solution | SD | ET |
|---|---|---|---|---|---|---|
| MIHA | BJ60_01 | 10,864.7334 | 10,914.4491 | 10,939.8273 | 21.2864 | 179.3093 |
|  | BJ60_02 | 10,217.4691 | 10,310.1949 | 10,395.6818 | 54.3404 | 167.1625 |
|  | BJ60_03 | 11,321.6848 | 11,423.8391 | 11,489.1644 | 53.3224 | 168.3861 |
|  | BJ60_04 | 11,800.7563 | 11,847.7284 | 11,915.0534 | 31.7880 | 179.6456 |
|  | BJ60_05 | 10,890.8377 | 10,909.8830 | 10,947.0117 | 14.5797 | 190.0507 |
|  | BJ60_06 | 11,875.3018 | 11,916.5380 | 11,942.8835 | 21.5401 | 157.5884 |
|  | BJ60_07 | 12,528.3080 | 12,634.5020 | 12,685.8953 | 54.0499 | 138.2126 |
|  | BJ60_08 | 11,783.3490 | 11,867.4716 | 11,932.7904 | 45.1478 | 162.5045 |
|  | BJ60_09 | 11,573.5410 | 11,705.3953 | 11,908.5312 | 107.6724 | 173.2149 |
|  | BJ60_10 | 12,939.9900 | 13,196.1938 | 13,269.0754 | 90.2917 | 161.0214 |
|  | *Average* | 11,579.5971 | 11,672.6195 | 11,742.5914 | 49.4019 | 167.7096 |
| MIHA$_{-level3}$ | BJ60_01 | 10,876.9472 | 10,929.2492 | 10,959.7533 | 23.8597 | 99.9533 |
|  | BJ60_02 | 10,354.6481 | 10,439.4020 | 10,529.0454 | 47.3483 | 98.3095 |
|  | BJ60_03 | 11,417.6826 | 11,510.4497 | 11,651.4713 | 77.5482 | 99.3993 |
|  | BJ60_04 | 11,810.6944 | 11,850.3646 | 11,947.2880 | 36.0053 | 104.0135 |
|  | BJ60_05 | 10,900.3834 | 10,930.1086 | 10,996.6298 | 29.6276 | 115.6167 |
|  | BJ60_06 | 11,884.6734 | 11,934.8017 | 11,994.1760 | 37.2390 | 100.8201 |
|  | BJ60_07 | 12,570.4124 | 12,737.4635 | 12,916.3772 | 116.3777 | 88.5942 |
|  | BJ60_08 | 11,816.6496 | 11,865.4873 | 11,937.1039 | 42.1579 | 94.7336 |
|  | BJ60_09 | 11,652.5596 | 11,744.5783 | 11,860.8162 | 74.2931 | 92.3574 |
|  | BJ60_10 | 13168.2194 | 13,240.3696 | 13,322.1267 | 43.8999 | 85.6219 |
|  | *Average* | 11,645.2870 | 11,718.2275 | 11,811.4788 | 52.8356 | 97.9418 |

**Table 6.** Comparison of MIHA and MIHA$_{-TS}$.

|  | Instance | Best Solution | Mean Solution | Worst Solution | SD | ET |
|---|---|---|---|---|---|---|
| MIHA | BJ60_01 | 10,864.7334 | 10,914.4491 | 10,939.8273 | 21.2864 | 179.3093 |
|  | BJ60_02 | 10,217.4691 | 10,310.1949 | 10,395.6818 | 54.3404 | 167.1625 |
|  | BJ60_03 | 11,321.6848 | 11,423.8391 | 11,489.1644 | 53.3224 | 168.3861 |
|  | BJ60_04 | 11,800.7563 | 11,847.7284 | 11,915.0534 | 31.7880 | 179.6456 |
|  | BJ60_05 | 10,890.8377 | 10,909.8830 | 10,947.0117 | 14.5797 | 190.0507 |
|  | BJ60_06 | 11,875.3018 | 11,916.5380 | 11,942.8835 | 21.5401 | 157.5884 |
|  | BJ60_07 | 12,528.3080 | 12,634.5020 | 12,685.8953 | 54.0499 | 138.2126 |
|  | BJ60_08 | 11,783.3490 | 11,867.4716 | 11,932.7904 | 45.1478 | 162.5045 |
|  | BJ60_09 | 11,573.5410 | 11,705.3953 | 11,908.5312 | 107.6724 | 173.2149 |
|  | BJ60_10 | 12,939.9900 | 13,196.1938 | 13,269.0754 | 90.2917 | 161.0214 |
|  | *Average* | 11,579.5971 | 11,672.6195 | 11,742.5914 | 49.4019 | 167.7096 |
| MIHA$_{-TS}$ | BJ60_01 | 10,887.9165 | 10,928.7764 | 10,958.6152 | 22.1538 | 179.1811 |
|  | BJ60_02 | 10,350.4335 | 10,433.3471 | 10,569.4033 | 66.3863 | 165.9635 |
|  | BJ60_03 | 11,410.8506 | 11,546.6879 | 11,693.6700 | 83.6334 | 183.0199 |
|  | BJ60_04 | 11,808.0504 | 11,876.0743 | 12,002.3412 | 60.3121 | 190.7105 |
|  | BJ60_05 | 10,893.5351 | 10,908.2957 | 10,941.6042 | 15.4682 | 213.1253 |
|  | BJ60_06 | 11,930.4763 | 11,993.7405 | 12,040.6174 | 29.2162 | 162.8483 |
|  | BJ60_07 | 12,790.4131 | 12,945.8023 | 13,093.2836 | 95.8863 | 150.9155 |
|  | BJ60_08 | 11,840.7762 | 11,980.0637 | 12,070.6884 | 64.6119 | 170.2074 |
|  | BJ60_09 | 11,700.6390 | 12,040.8509 | 12,195.6716 | 140.5182 | 169.8237 |
|  | BJ60_10 | 13,352.3191 | 13,452.7316 | 13,528.2388 | 54.9095 | 152.6279 |
|  | *Average* | 11,696.5410 | 11,810.6370 | 11,909.4134 | 63.3096 | 173.8423 |

**Table 7.** Comparison of MIHA and MIHA$_{-GA}$.

|  | Instance | Best Solution | Mean Solution | Worst Solution | SD | ET |
|---|---|---|---|---|---|---|
| MIHA | BJ60_01 | 10,864.7334 | 10,914.4491 | 10,939.8273 | 21.2864 | 179.3093 |
|  | BJ60_02 | 10,217.4691 | 10,310.1949 | 10,395.6818 | 54.3404 | 167.1625 |
|  | BJ60_03 | 11,321.6848 | 11,423.8391 | 11,489.1644 | 53.3224 | 168.3861 |
|  | BJ60_04 | 11,800.7563 | 11,847.7284 | 11,915.0534 | 31.7880 | 179.6456 |
|  | BJ60_05 | 10,890.8377 | 10,909.8830 | 10,947.0117 | 14.5797 | 190.0507 |
|  | BJ60_06 | 11,875.3018 | 11,916.5380 | 11,942.8835 | 21.5401 | 157.5884 |
|  | BJ60_07 | 12,528.3080 | 12,634.5020 | 12,685.8953 | 54.0499 | 138.2126 |
|  | BJ60_08 | 11,783.3490 | 11,867.4716 | 11,932.7904 | 45.1478 | 162.5045 |
|  | BJ60_09 | 11,573.5410 | 11,705.3953 | 11,908.5312 | 107.6724 | 173.2149 |
|  | BJ60_10 | 12,939.9900 | 13,196.1938 | 13,269.0754 | 90.2917 | 161.0214 |
|  | *Average* | 11,579.5971 | 11,672.6195 | 11,742.5914 | 49.4019 | 167.7096 |
| MIHA$_{-GA}$ | BJ60_01 | 10,889.6733 | 10,920.5130 | 10,947.4318 | 20.3041 | 162.6233 |
|  | BJ60_02 | 10,267.1195 | 10,333.7142 | 10,415.6234 | 49.4848 | 149.6058 |
|  | BJ60_03 | 11,349.3097 | 11,473.1526 | 11,595.0291 | 77.7908 | 149.0490 |
|  | BJ60_04 | 11,804.8311 | 11,843.4759 | 11,937.9748 | 39.7545 | 150.1856 |
|  | BJ60_05 | 10,898.5533 | 10,917.0069 | 10,942.3023 | 14.3160 | 168.1142 |
|  | BJ60_06 | 11,882.7327 | 11,939.8357 | 11,973.6663 | 30.1972 | 170.9655 |
|  | BJ60_07 | 12,534.6436 | 12,658.8725 | 12,849.5941 | 85.0550 | 129.9516 |
|  | BJ60_08 | 11,806.8383 | 11,876.4740 | 11,934.8815 | 39.6358 | 138.7723 |
|  | BJ60_09 | 11,630.3241 | 11,687.7664 | 11,850.7900 | 63.9084 | 140.8432 |
|  | BJ60_10 | 13,120.3543 | 13,262.1483 | 13,355.0285 | 75.4520 | 128.9989 |
|  | *Average* | 11,618.4380 | 11,691.2960 | 11,780.2322 | 49.5899 | 148.9109 |

*5.5. Effectiveness of the Membrane Structure*

In this subsection, we verify the effectiveness of the membrane structure. Generally, membrane computing provides a parallel distributed framework for solving the optimisation problem. We denoted our algorithm without the membrane structure as MIHA$_{-MS}$, in which all membranes were removed except for a single level-2 membrane, and the output was the best solution obtained by the algorithm in it. As shown in Table 8, without the membrane framework, the MIHA$_{-MS}$ was far less effective in finding competitive solutions in comparison with the results obtained by using MIHA.

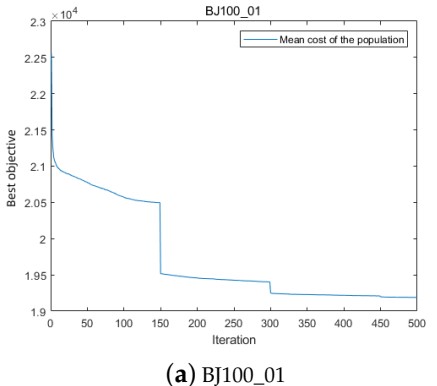

**(a)** BJ100_01

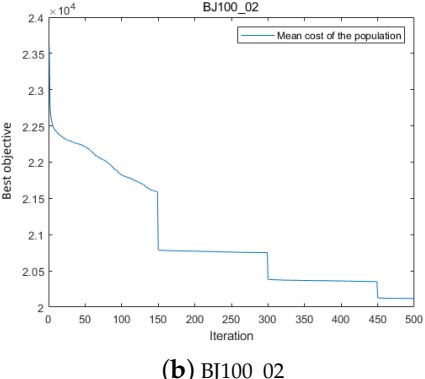

**(b)** BJ100_02

**Figure 6.** *Cont.*

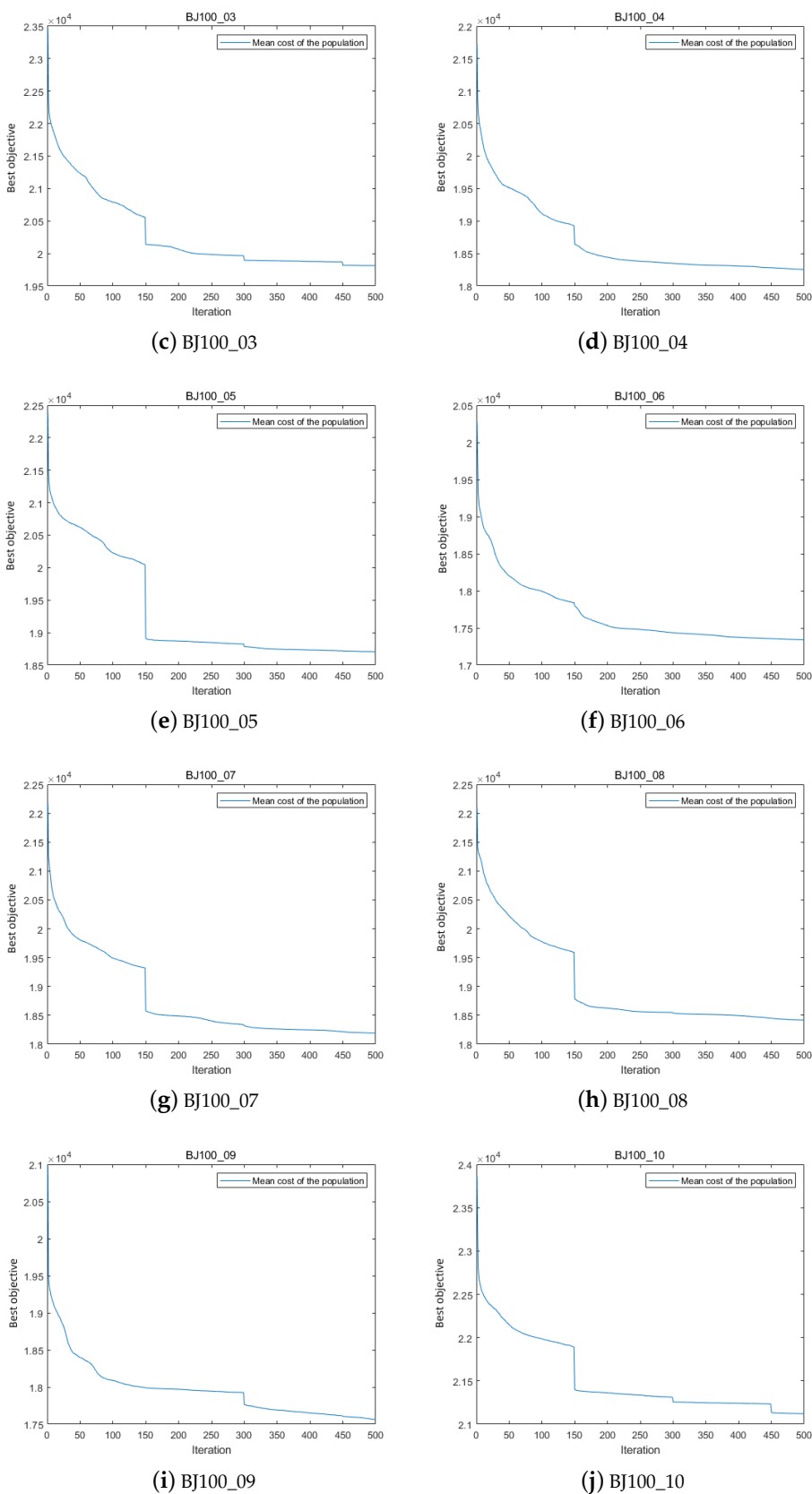

**Figure 6.** Convergence process of GA in skin membrane. (Cliffs are observed every $I_{MCA}$ iterations in the convergence curves, which proves the effectiveness of the MCA.)

**Table 8.** Comparison of the MIHA and MIHA$_{-MS}$.

|  | Instance | Best Solution | Mean Solution | Worst Solution | SD | ET |
|---|---|---|---|---|---|---|
| MIHA | BJ60_01 | 10,864.7334 | 10,914.4491 | 10,939.8273 | 21.2864 | 179.3093 |
|  | BJ60_02 | 10,217.4691 | 10,310.1949 | 10,395.6818 | 54.3404 | 167.1625 |
|  | BJ60_03 | 11,321.6848 | 11,423.8391 | 11,489.1644 | 53.3224 | 168.3861 |
|  | BJ60_04 | 11,800.7563 | 11,847.7284 | 11,915.0534 | 31.7880 | 179.6456 |
|  | BJ60_05 | 10,890.8377 | 10,909.8830 | 10,947.0117 | 14.5797 | 190.0507 |
|  | BJ60_06 | 11,875.3018 | 11,916.5380 | 11,942.8835 | 21.5401 | 157.5884 |
|  | BJ60_07 | 12,528.3080 | 12,634.5020 | 12,685.8953 | 54.0499 | 138.2126 |
|  | BJ60_08 | 11,783.3490 | 11,867.4716 | 11,932.7904 | 45.1478 | 162.5045 |
|  | BJ60_09 | 11,573.5410 | 11,705.3953 | 11,908.5312 | 107.6724 | 173.2149 |
|  | BJ60_10 | 12,939.9900 | 13,196.1938 | 13,269.0754 | 90.2917 | 161.0214 |
|  | *Average* | 11,579.5971 | 11,672.6195 | 11,742.5914 | 49.4019 | 167.7096 |
| MIHA$_{-MS}$ | BJ60_01 | 11,328.7061 | 11,450.7661 | 11,645.5626 | 102.4337 | 16.3996 |
|  | BJ60_02 | 10,706.7220 | 10,825.8266 | 10,952.4319 | 77.1375 | 16.0414 |
|  | BJ60_03 | 12,036.9001 | 12,135.7632 | 12,194.5568 | 54.3717 | 15.1175 |
|  | BJ60_04 | 12,367.3172 | 12,561.4857 | 12,854.7533 | 147.9147 | 17.2651 |
|  | BJ60_05 | 11,232.0918 | 11,294.0261 | 11,371.0893 | 52.7390 | 16.8575 |
|  | BJ60_06 | 12,288.4131 | 12,446.6655 | 12,589.8602 | 89.2907 | 16.7956 |
|  | BJ60_07 | 13,243.3081 | 13,393.8310 | 13,479.3281 | 84.1279 | 13.8604 |
|  | BJ60_08 | 12,172.4397 | 12,363.8495 | 12,507.0350 | 110.1176 | 13.1230 |
|  | BJ60_09 | 12,293.5838 | 12,422.8764 | 12,574.3870 | 75.6798 | 14.9571 |
|  | BJ60_10 | 13,356.4703 | 13,585.7384 | 13,783.8378 | 130.1142 | 13.2258 |
|  | *Average* | 12,102.5952 | 12,248.0829 | 12,395.2842 | 92.3927 | 15.3643 |

### 5.6. Computational Result of Larger-Scale Problems

In this subsection, to further demonstrate the practical property of MIHA, the experiments were conducted on the larger-scale real-world problem instances, i.e., the Beijing 100-nodes set and the Jingdong 1000-nodes instance, respectively, where our method was set with $p_{level3}$ = 0.8 and $I_{MCA}$ = 150. The results of the Beijing 100-nodes set are recorded in Table 9, and the results of the Jingdong 1000-nodes instance are recorded in Table 10. The computational result verified the favourable capability and superiority of the proposed MIHA in solving the real-world large-scale problem. In particular, in the comparison of different instance sizes, as shown in Table 11, a roughly linearly growing ET was observed, which justified the advantage of our algorithm in solving the large-scale problems.

**Table 9.** Computational result of the Beijing 100-node problem.

| Instance | Best Solution | Mean Solution | Worst Solution | SD | ET |
|---|---|---|---|---|---|
| BJ100_01 | 19,067.0291 | 19,198.1247 | 19,286.2464 | 77.2545 | 177.5238 |
| BJ100_02 | 20,054.6213 | 20,212.3116 | 20,335.2703 | 79.4795 | 177.5122 |
| BJ100_03 | 19,611.4194 | 19,739.1057 | 19,807.4854 | 69.5798 | 187.4094 |
| BJ100_04 | 18,185.2382 | 18,288.2393 | 18,365.8098 | 66.8545 | 183.9430 |
| BJ100_05 | 18,450.2424 | 18,650.0880 | 18,721.6565 | 78.3719 | 177.8560 |
| BJ100_06 | 17,185.9916 | 17,292.5207 | 17,429.9383 | 100.3186 | 191.1868 |
| BJ100_07 | 18,018.7958 | 18,130.9018 | 18,231.9109 | 58.4036 | 203.7468 |
| BJ100_08 | 18,077.4404 | 18,245.1333 | 18,386.8610 | 102.6939 | 190.7556 |
| BJ100_09 | 17,317.3479 | 17,414.3330 | 17,513.4761 | 57.2721 | 181.7808 |
| BJ100_10 | 20,888.4245 | 20,996.4199 | 21,191.1938 | 87.1091 | 165.4292 |
| *Average* | 18,685.6551 | 18,816.7178 | 18,926.9849 | 77.7338 | 183.7144 |

**Table 10.** Computational result of the Jingdong 1000-node problem.

| Algorithm | Best Solution | Mean Solution | Worst Solution | SD | ET |
|---|---|---|---|---|---|
| MIHA | 140,577.0284 | 141,085.6778 | 141,823.7589 | 365.4070 | 2331.9363 |

**Table 11.** The mean elapsed time of different instance sizes.

| Instance | BJ60 | BJ100 | JD1000 |
|:---:|:---:|:---:|:---:|
| ET | 150.6687 | 183.7144 | 2331.9363 |

### 5.7. Comparison with Other Algorithms

Three heuristic algorithms, i.e., the hybrid tabu search [17], the tabu search, and the GA [33] were implemented as baselines to compare with our MIHA, the results of which are recorded in Figures 7 and 8. From Figure 7, we see that MIHA outperformed the hybrid tabu search, the tabu search, and the GA in terms of solution quality and convergence speed for all ten instances of BJ100, respectively. Moreover, Figure 8 shows the results of the four algorithms for the JD1000 instance, where our MIHA not only achieved the lowest cost but also presented the highest convergence speed. This further demonstrates the significant advantage of MIHA in solving a large-scale problem.

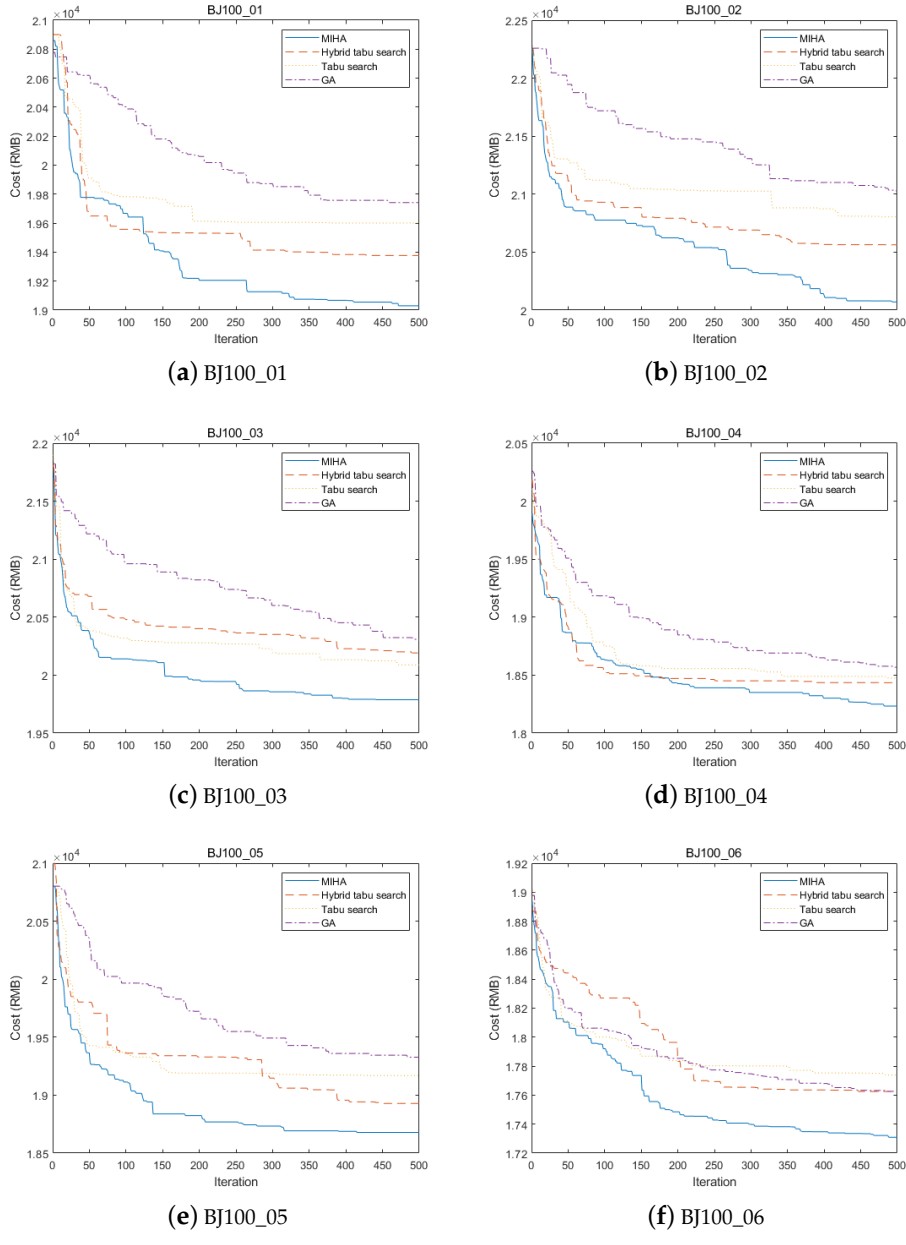

(**a**) BJ100_01

(**b**) BJ100_02

(**c**) BJ100_03

(**d**) BJ100_04

(**e**) BJ100_05

(**f**) BJ100_06

**Figure 7.** *Cont.*

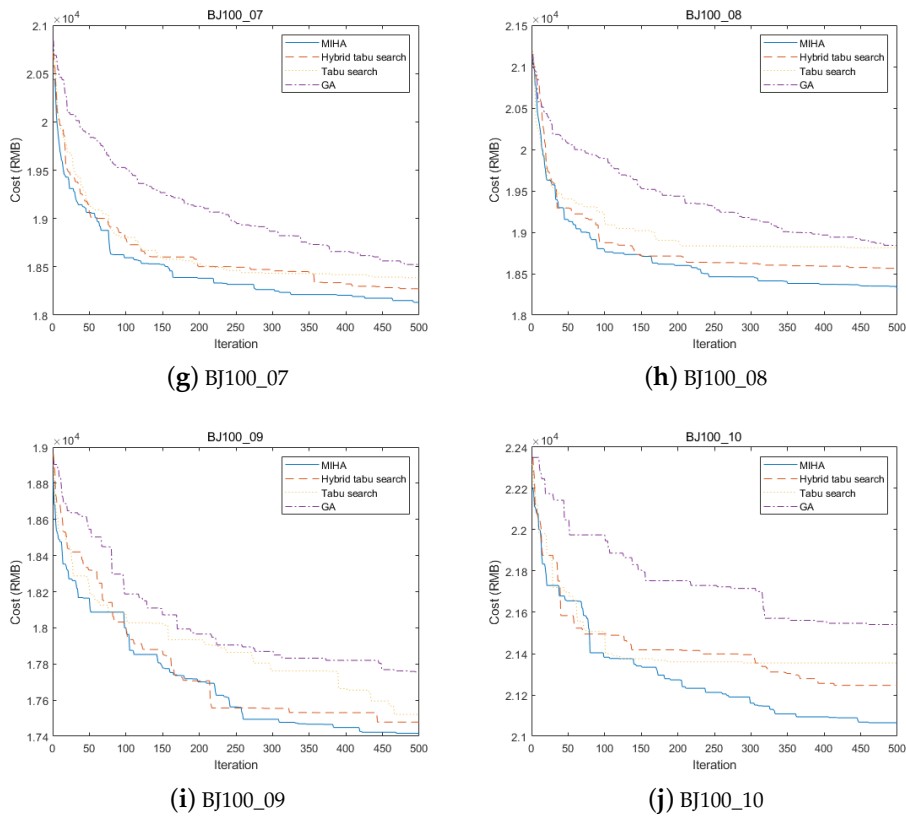

**Figure 7.** The convergence curves of the MIHA, hybrid Tabu search, Tabu search, and GA for 10 instances of BJ100. (The MIHA outperforms other algorithms in terms of solution quality and convergence speed in all instances of BJ100.)

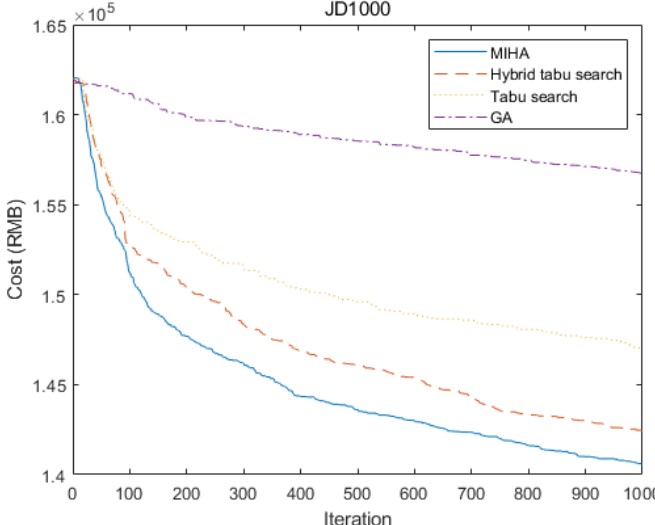

**Figure 8.** The convergence curves of MIHA, hybrid tabu search, tabu search, and GA for the JD1000 problem instance.

## 6. Conclusions

In this paper, we designed a membrane-inspired framework to improve the performance of the heuristics when dealing with a realistic and large-scale green open vehicle routing problem with time windows. The proposed method, i.e., a membrane-inspired hybrid algorithm, benefits from the parallel distributed structure and a unique communication strategy in the P system. The computational results based on the Beijing dataset and

Jingdong instance justified its strong competitiveness against other baselines, where our algorithm achieved the lowest overall cost, which comprises fuel cost, emission cost, and driver cost, on all tested instances.

The green open vehicle routing problem with time windows, as a notable variant of the vehicle routing problem, has considerable value in the vigorous development of the sharing economy, green logistics, and sustainable society. In the future, more realistic models involving the green open routes will be investigated, such as the green close–open vehicle routing problem and variants that consider a heterogeneous fleet. Our algorithm might still be able to solve them with appropriate modifications, given the desirable generalisation capability of the membrane-inspired algorithms for solving hard optimisation problems. We also plan to integrate our work with the deep (reinforcement)-learning-based methods developed by Li et al. [49], Wu et al. [50], and Xin et al. [51] for solving routing problems, so that it allows the membrane-inspired algorithms to be more intelligent. In addition, the engine of the vehicle poses many advantages to improve the performance and parameter characteristics [52], and carbon emissions can be reduced if an appropriate injection strategy is adopted [53]. Therefore, the improvement of the engine should be considered in future work. Future research directions can also focus on developing more optimisation methods such as the lion optimisation algorithm [54], red deer algorithm [55], etc.

**Author Contributions:** Conceptualization, Y.N. and Z.Y.; methodology, Y.N., Z.Y and J.X.; writing—original draft preparation, Z.Y.; writing—review and editing, R.W.; supervision, S.Z.; funding acquisition, Y.N. and J.X. All authors have read and agreed to the published version of the manuscript.

**Funding:** This work was supported by the National Natural Science Foundation of China [grant numbers 62172373, 61872325, 62072258, 61772290] and the Fundamental Research Funds for the Central Universities, China [grant number 2652019028].

**Institutional Review Board Statement:** Not applicable.

**Informed Consent Statement:** Not applicable.

**Data Availability Statement:** The data presented in this study are available on request from the corresponding author.

**Conflicts of Interest:** The authors declare no conflict of interest.

## Notations

| | |
|---|---|
| $G$ | Complete directed graph |
| $N$ | Node set |
| $N_0$ | Customer set |
| $A$ | Arc set |
| $Q$ | Vehicle capacity |
| $w$ | The weight of a vehicle |
| $q_i$ | The demand of customer $i$ |
| $s_i$ | The time cost of the route ending with $i$ |
| $y_i$ | The actual starting time for serving node $i$ |
| $d_{ij}$ | The length of arc $(i, j)$ |
| $f_{ij}$ | The amount of freight flow on arc $(i, j)$ |
| $x_{ij}$ | Binary flag variable 1 |
| $z_{ij}^r$ | Binary flag variable 2 |
| $v_{ij}^r$ | The travel speed of vehicle $r$ on arc $(i, j)$ |
| $[a_i, b_i]$ | The time window of customer $i$ |

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
