# Peer review of "Solving the Green Open Vehicle Routing Problem Using a Membrane-Inspired Hybrid Algorithm"

_sustainability, doi:10.3390/su14148661_

Round 1
Reviewer 1 Report
In the manuscript of sustainability-1791412, the authors design a membrane-inspired framework to improve the performance of the heuristics when dealing with realistic and large-scale GOVRPTW. The proposed method, i.e., MIHA, benefits from the parallel distributed structure and a unique communication strategy in the P system. The computational results based on the Beijing dataset and Jingdong instance justified its strong competitiveness against other baselines, where our MIHA achieves the lowest overall cost, which comprises fuel cost, emission cost, and driver cost, on all tested instances. Overall, the topic is very interesting and the article seems full of content. Moreover, it will be appropriate to consider the following corrections and comments.
1. Please also avoid "lump sum references", such as XXXXX [1-5]; all references should be cited with a detailed and specific description. In the references, all authors should be included, avoiding using "et. al.";
2. The car's engine promise many advantages to improve car's performance and parameter characteristics. For clarification, the authors should refer to Potential improvement in combustion and pollutant emissions of a hydrogen-enriched rotary engine by using novel recess configuration; Parametric analysis of hydrogen two-stage direct-injection on combustion characteristics, knock propensity, and emissions formation in a rotary engine.
3. The selection of vehicle speed in 3.6 is based on the minimum cost and driver's salary, but in practice the vehicle speed is often determined based on the road speed limit, and drivers tend to use the maximum speed to make round trips between nodes, has it been considered that the vehicle speed is determined based on the road as well as the driver when selecting the vehicle speed.
4. Please avoid using abbreviations in the TITLE, HIGHLIGHTS, ABSTRACT and CONCLUSION if possible.
5. CAPTIONS: Captions for figures and tables should be presented with more specific description rather than a general sentence like "Results of the experiments ...", "A studied system ...."
6. An updated and complete literature review should be conducted to present the state-of-the-art and knowledge gaps of the research with strong relevance to the topic of the paper.
Reviewer 2 Report
The manuscript should be improved according to the below comments.
The problem description is missing. Prior to the problem formulation, the problem description in words should be provided, so that readers can clearly understand the problem under consideration. The assumptions should be explicitly stated. The following assumptions are assumed in this article but never clearly stated: (i) the goods delivery is considered as opposed to the goods pick-up problem, (ii) demand at any customer qi is assumed less than the vehicle capacity Q, and (iii) the traffic conditions on all roads are uncongested or free-flow such that the vehicle speeds can be optimized.
Detailed Comments
· In the abstract, the level-2 membranes are referred to as adjacent outer membranes on line 9 and line membranes on line 12. This can confuse readers. Please be consistent.
· On line 11, “a new variant of the VRPTW” should read “a variant of the VRPTW”.
· The phrase “In specific, ” is used throughout the manuscript. This should read “Specifically,” or “To be specific,”.
· N is defined as the set of nodes on line 97, and N is defined as engine speed in Table 1. The same notation should not be defined differently in the same manuscript.
· On line 113, the notation R appears for the first time and it is never defined in the article.
· On line 113-114, the objective is described as the sum of three components, but in fact, it is composed of four components. The fourth component is not described.
· The notations lambda, fc, gamma, alpha, N, V, beta, fd and sj should be defined in the paragraph after Equation (4), but they are defined on line 138. This makes the article more difficult to follow.
· On lines 118-119, notations tau, theta, Cr, g, Cd, rho, A, K, nt, f, eta, psi, xi, appear for the first time without their descriptions. This makes the article more difficult to follow.
· On line 135, y0i is defined as the departure time from the depot and the first visited customer is j. This is not used in the formulation as it does not appear in Equations (1)-(17).
· On line 138, sj is defined as the total time cost of the route in which the last visited customer is j. sj appears in the objective function. sj does not appear in any constraint. Since sj is a decision variable, there must be an equation to determine sj. Thus, the current formulation is not complete.
· On line 143, in the equation “Mij=max{0, bi-si+dij/vr-aj}”, si should be replaced by ti.
· In Table 1, some units should spell out for clarity such as L, kj, rev.
· On lines 175-176, the complete solution consists of a number of routes and the vehicle-length chromosomes comprising the integer nodes representing a route. The service start time yi and selected speed zrij are not mentioned in the solution encoding.
· On line 181, “In the skin membrane, six unique operators are adopted to create different initial solutions” indicates that the level-1 membrane and level-2 membrane are communicated through bidirectional channels as opposed to unidirectional channels. Please clarify.
· In Figure 2, “insert-1” should read “insert-I1”.
· On line 211, the insert I1 should be briefly described in words.
· In Equation (21), yju is used without its definition.
· In Equation (18), “optimum” implies “minimum” or “maximum”. This is unclear.
· Algorithm 2 is not referred to in the text.
· In Figure 4, “the adjacent membrane” and “the former membrane” are employed. These should be replaced with the specific names e.g. level-2 membranes and level-1 membranes.
· The table summarizing the description of notations should be added.
· Equations (1)-(4) are in fact the objective function. Equations (2)-(4) are not equations as they are just the second term to the fourth term in the objective function.
· On line 298, “attractors” are used. Please describe the meaning of this term.
· In Algorithm 5, “that solutions” should read “those solutions”.
· In section 4.1.1, plevel3 is analyzed. However, the parameter plevel3 is never mentioned in the algorithm.
· On line 372, the authors should briefly describe the MIHA-GA and how it differs from MIHA.
· In Figure 6, the vertical axis label should be the best objective as opposed to “cost”.
· On line 384, the authors should briefly describe the MIHA-MS and how it differs from MIHA.
· The computer specification that used in the experiment is missing. The CPU time for all the runs are missing. The comparison of algorithmic performance include the solution quality, convergence speed and CPU time.
· On lines 395-397, the authors compare the elapsed time (ET) of 1000-node instance and the ET of 100-node instance, and conclude that the ET is linearly growing. This is incorrect because any two data points will yield a linear trend. The authors should study the ET of different instance size, e.g., 100 nodes, 200 nodes, 300 nodes,…, 1000 nodes, and check whether the relationship of the ET and the instance size is linear or polynomial or exponential.
· On line 418, “an notable” should read “a notable”.
Reviewer 3 Report
The proposed solution method is very messy. It is very hard to follow the structure of the proposed method. The authors should explain different steps of the proposed method in a systematic way.
Although there are many methods in the literature for solving similar problems, the authors didn’t compare the proposed method with them.
Figures 3, 4 and 5 are a bit confusing. The authors can use pseudo code instead of them.
To prove the performance of the proposed method, a comprehensive statistical test is required.
Limitations of the study should be discussed and future directions should be outlined for example, using other optimisation methods such as the Lion Optimisation algorithm or Red deer algorithm.
Round 2
Reviewer 1 Report
The authors strictly modified the manuscript according to the reviews' comments and suggestions. Thus the current version is ready to be accepted.
Author Response
Thanks for your valuable suggestions.
Reviewer 3 Report
The paper has been improved significantly. Only in line 505 please support the sentence using appropriate reference.
Best wishes.
